# Sediment Assessment of the Pchelina Reservoir, Bulgaria

**DOI:** 10.3390/molecules26247517

**Published:** 2021-12-11

**Authors:** Tony Venelinov, Veronika Mihaylova, Rositsa Peycheva, Miroslav Todorov, Galina Yotova, Boyan Todorov, Valentina Lyubomirova, Stefan Tsakovski

**Affiliations:** 1Chair of Water Supply, Sewerage, Water and Wastewater Treatment, Faculty of Hydraulic Engineering, University of Architecture, Civil Engineering and Geodesy, 1 Hr. Smirnenski Blvd., 1046 Sofia, Bulgaria; tvenelinov_fhe@uacg.bg; 2Chair of Analytical Chemistry, Faculty of Chemistry and Pharmacy, Sofia University “St. Kliment Ohridski”, 1 J. Bourchier Blvd., 1164 Sofia, Bulgaria; v.mihaylova@chem.uni-sofia.bg (V.M.); g.yotova@chem.uni-sofia.bg (G.Y.); b.todorov@chem.uni-sofia.bg (B.T.); vlah@chem.uni-sofia.bg (V.L.); 3DIAL Ltd., 111 Mina Buhovo Str., 1830 Sofia-Buhovo, Bulgaria; rbarganska11@yahoo.bg; 4Chair of Hydrotechnics, Faculty of Transportation Engineering, University of Architecture, Civil Engineering and Geodesy, 1 Hr. Smirnenski Blvd., 1046 Sofia, Bulgaria; miro_todorof@yahoo.com

**Keywords:** Pchelina Reservoir, sediment, principal component analysis, Mann–Kendall test, ecotoxicity

## Abstract

The temporal dynamics of anthropogenic impacts on the Pchelina Reservoir is assessed based on chemical element analysis of three sediment cores at a depth of about 100–130 cm below the surface water. The ^137^Cs activity is measured to identify the layers corresponding to the 1986 Chernobyl accident. The obtained dating of sediment cores gives an average sedimentation rate of 0.44 cm/year in the Pchelina Reservoir. The elements’ depth profiles (Ti, Mn, Fe, Zn, Cr, Ni, Cu, Mo, Sn, Sb, Pb, Co, Cd, Ce, Tl, Bi, Gd, La, Th and U_nat_) outline the Struma River as the main anthropogenic source for Pchelina Reservoir sediments. The principal component analysis reveals two groups of chemical elements connected with the anthropogenic impacts. The first group of chemical elements (Mn, Fe, Cr, Ni, Cu, Mo, Sn, Sb and Co) has increasing time trends in the Struma sediment core and no trend or decreasing ones at the Pchelina sampling core. The behavior of these elements is determined by the change of the profile of the industry in the Pernik town during the 1990s. The second group of elements (Zn, Pb, Cd, Bi and U_nat_) has increasing time trends in Struma and Pchelina sediment cores. The increased concentrations of these elements during the whole investigated period have led to moderate enrichments for Pb and U_nat_, and significant enrichments for Zn and Cd at the Pchelina sampling site. The moderately contaminated, according to the geoaccumulation indexes, Pchelina Reservoir surface sediment samples have low ecotoxicity.

## 1. Introduction

Chemical elements are among the most widespread of the various pollutants originating from anthropogenic activities, particularly from mining, metallurgy and smelting waste sites. They are one of the most persistent pollutants in the environment, since they do not decompose, nor do they biodegrade into simpler and less harmful substances. Their concentrations in water are quite variable, with the highest concentrations found in the suspended matter (insoluble substances) and the lowest in the liquid phase [1,2,3]. The distribution of the elements is primarily on the surfaces of sediments, suspended particles, and other solids, decreasing in the order: suspended matter > sediment > water [4,5]. Chemical processes occurring in natural waters, such as oxidation, can cause them to precipitate from the solution [6].

As an integral part of natural water sources, sediments play an important role in the biogeochemical cycle of the elements, as they are the site of deposition and chemical transformation of many compounds entering the water bodies [7]. The elements enter surface waters from many sources, in the form of atmospheric deposits, or are leached from rocks and soil. They are not biodegradable, but bind to proteins, thus being stored in the bodies of water organisms or excreted in their feces [8], which under certain conditions leads to secondary contamination of the water bodies. In natural waters, chemical elements are found in many forms: as free ions (the most toxic forms for living organisms); in the form of various complexes; as precipitated compounds suspended in the aqueous phase; and adsorbed on the surface of other suspended or colloidal particles [9].

The history of anthropogenic water pollution can be determined by analysis of sediments [10,11,12]. Reservoir sediments provide fine-scale information on the historical record of metal pollution in a watershed [13,14,15]. They have recorded the elemental deposition and thus allow establishing a connection between the temporal evolution of the pollution and historical changes in smelting and waste treatment processes [13]. The resulting compositional datasets are usually tested by principal component analyses, self-organizing maps, and cluster analyses, with their pollution load index (PLI), index of geoaccumulation (*I_geo_*) and enrichment factors (EFs) being calculated [16,17,18,19,20,21,22,23,24,25,26].

Reservoir sediments accumulate at a high rate (usually > 2 cm/year [27]), in contrast to river sediments (usually < 0.3 cm/year [26]), which is why they can reveal the accumulation of the elements over time. Due to these sedimentation rates, reservoir sediments are considered to be slightly affected by early diagenesis processes and provide preserved historical elemental inputs. Reservoir sediments are also of great concern, since they can turn from a sink to a source of chemical elements for fluvial systems by diffusion at the water/sediment interface, bioturbation or resuspension due to dredging or flooding. Thus, it is important to determine the intensity of pollution by inventorying the elemental concentrations and their spatial distribution in sediments [28].

Pchelina Reservoir was built in 1975 and serves not only as a secondary precipitator of the Struma River but also as a source for irrigation and industrial water supply of Pernik Municipality, Bulgaria. The total volume of the reservoir is 54.8 million m^3^ (of which the useful volume is 19.3 million m^3^), and its depth reaches 19 m [29]. The main sources of water pollution at Pchelina Reservoir are defined as point and diffuse. Point sources are the sewerage systems from the settlements, discharged without purification in treatment plants, the wastewater treatment plants, the industrial sites, the tailings, and the mines. Diffuse sources of pollution are unregulated landfills for solid waste, settlements without sewerage, landfills, and agricultural activities such as animal husbandry. The main pollutant of the river Struma is the town of Pernik, the heavy industry of which has traditionally been dominated by mining activities and metallurgy. The industrial profile of the town changed in the 1990s, but the wastewaters discharged into the river are still collected in the Pchelina Reservoir. The only available studies of the sediments of Pchelina Reservoir were made by Meuser and co-authors [30,31].

This study aims to propose a methodology for assessing the temporal dynamics of anthropogenic impacts on sediments of Pchelina Reservoir. The proposed methodology includes: (i) analysis of chemical elements and ^137^Cs radionuclides of three sediment cores taken from three selected sampling points in Pchelina Reservoir, (ii) multivariate and time trend statistical analysis of the sediment cores data, and (iii) calculation of enrichment factors and geochemical indexes of surface sediments followed by ecotoxicity assessment using Phytotoxkit F ™ bioassay.

## 2. Results

To reveal the relationships between the analyzed chemical elements and/or layers in the sediment cores, a principal component analysis (PCA) was applied. The input data set used for PCA consists of 58 objects (layers in the sedimentary cores) and 20 variables (analyzed chemical elements). PCA of the data from the three sedimentary cores revealed that the first three main components describe almost 80% of the variation of the data. The number of latent variables is determined based on their eigenvalues and the internal model validation error. In the formation of the first principal component, explaining 41.53% of data variance, the following elements have a significant impact: Mn, Fe, Cr, Ni, Cu, Mo, Sn, Sb and Co (Figure 1). This component separates the elements into two groups according to their time trends in Struma sediment core: (i) Mn, Fe, Zn, Cr, Ni, Cu, Mo, Sn, Sb, Pb, Co, Cd and U have increased concentrations with time; and (ii) Ti, Ce, Tl, Bi, Gd, La and Th have decreased or non-significant time trends.

The second principal component (22.16%) is formed by Ce, Gd, La, Th, and Ti, which significantly decrease over time in the sedimentary cores at Struma and Svetlia rivers (Figure 2). These decreasing trends do not lead to a significant change in the contents of the elements in the Pchelina Reservoir, which are comparable to the contents in the sedimentary cores near the Struma and Svetlia at the beginning of the period.

The third principal component (15.44%) is formed by the elements Zn, Pb, Cd, Bi and U_nat_. The factor scores of the layers of the Pchelina core show a particularly pronounced positive trend, which leads to the formation of two groups of layers. The first group, covering the beginning of the studied period (1–11), has contents comparable to the sedimentary core of the Svetlia (anthropogenically undisturbed river), while the contents of the elements in the second group (12–23) are the highest for all the three studied sediment cores (Figure 3). The layers in the Struma sediment core have medium factor scores between both groups of Pchelina sediment core.

^137^Cs has been widely applied as an environmental tracer in the study of sediment recent deposition history (usually within the last 50 years) [32,33]. Each of the 2 cm sediment core fractions was analyzed for its ^137^Cs content. Only in one of these fractions for each sediment core, radioactivity (γ-activity) was found and the content of ^137^Cs measured was between 32.8 Bq/kg to 55.3 Bq/kg specific activities. Based on these findings, a conclusion was drawn that, on average, 15 cm of sediment was deposited in the 34 years since the Chernobyl incident (1986). This means that the average sedimentation rate at the sampling points in Pchelina Reservoir is 0.44 cm/year. Such results differ from the literature values—usually > 2 cm/year [27,34]—but are closer to the reported values for the rates of river sediment—usually < 0.3 cm/year) [30].

The element depth profiles of sediments from Pchelina Reservoir at the different sampling points (Pchelina, Struma and Svetlia) are shown in Figure 4. Red points indicate the sample in which the highest radioactivity (γ-activity) has been registered, which corresponds to the Chernobyl pollution of 1986. A similar approach was used by Audry et al. [13] for the sediment core dating of the Lot River reservoirs.

To determine whether significant time trends of elements were observed, the Mann–Kendell test was performed. The results are presented in Table 1, with significant trends (*p* < 0.05) marked with “+” for increasing and “−” for decreasing trends. The significant time tendencies in the sediment cores influenced by the flowing rivers reveal that Struma River (sampling point 1) is the main source of Pchelina Reservoir pollution.

According to the sedimentation rate calculations, it is assumed that any concentration of each element in a layer more than 6 cm below the sample, which corresponds to the Chernobyl pollution (marked in red in Figure 4), is a background concentration. The average result for Fe (used as a conservative element) of all such samples was calculated for each sampling point and used to obtain the enrichment factor for the sample corresponding to 2020 (top layers, Table 2).

Significant enrichment is observed only in the reference point (Pchelina), in terms of Zn and Cd (7.6 and 7.5, respectively). Regarding Cd, the enrichment in Svetlia is moderate (3.2). The enrichment factor is 1.8 in the Struma, which shows significant pollution with this metal in the entire Reservoir. Except for these elements, moderate enrichment is observed for Cu, Pb, Ce and Th only at Pchelina, as well as for Tl and U_nat_ in Svetlia and Pchelina.

The same approach for determination of the reference concentration (average of all results for the layers more than 6 cm below the sample, which corresponds to the Chernobyl pollution—marked in red in Figure 4) for each of the studied elements was used to determine the geoaccumulation index. The results are presented in Table 3.

It is evident from Table 3 that for most elements, the geoaccumulation index corresponds to an uncontaminated sample. This result is especially important for the sediments at the inflow of the Svetlia River (sampling point 2), where only for Cd, Tl and U_nat_ the index corresponds to an unpolluted to moderately polluted sample. The geoaccumulation index in the sediments at the inflow of the Struma River (sampling point 1) corresponds to an unpolluted to moderately contaminated sample for Fe, Cr, Mo and Sn and to a moderately polluted sample for Sb and Cd. The geoaccumulation index for the sediments of Pchelina Reservoir from the village of Radibosh (sampling point 3), which corresponds to an unpolluted to moderately polluted sample in terms of Pb, Tl and U_nat_ (similar to the sediments in Svetlia) and a moderately polluted sample in terms of Zn and Cd (analogous to the sediments in the Struma). The moderate contamination of the sediments of Pchelina Reservoir in all sampling points with Cd is impressive. Similar results were obtained for sediments of Wadi Al-Arab Dam, Jordan [35], where sediments were found uncontaminated with Mn, Fe, and Cu, moderately contaminated with Zn, and strongly to extremely contaminated with Cd. Ye et al. [36] compared deposits of “heavy metals”, accumulated in the water-level-fluctuation zone before and after the submergence period and found that Cd is the main pollutant of the sediment.

The results of the conducted biotest Phytotoxkit F ™ are presented in Table 4 and reveal the low ecotoxicity of the surface sediments from Pchelina Reservoir.

The indicator related to seed germination (SG) of Sinapis alba shows the lack of ecotoxicological effect. This means that the number of germinated seeds in the analyzed sediments is equal to their number in the control sample, in this case, all 10 seeds have germinated. The ecotoxicological effect, reflecting the growth of the roots of Sinapis alba, is relatively weak, as it is highest in the sample from Struma River, and even shows a negative value in Pchelina (hormesis). This means that the roots of the test species used are longer than the control sample.

## 3. Discussion

The background concentrations strongly depend on geological characteristics such as mineral composition, grain size distribution and organic matter content. Thus, establishing geochemical background concentrations of chemical elements is a very important step in environmental pollution assessment [37]. ^137^Cs as an indicator of sedimentation processes is consistent as it binds almost irreversibly to clay and silt particles and because of its relatively long half-life (*t_1/2_* = 30.2 years). Moreover, the Chernobyl nuclear accident of 1986 has been recorded by the European sediments. Thus, ^137^Cs activity depth profiles are often used for sediment core dating [38]. 

Based on the measurement of γ-activity and the calculated sedimentation rate in the Pchelina Reservoir of an average of 0.44 cm/year, it can be assumed that three of the points under 1986 (marked in red) are samples of sediment (approximately 6 cm), and the rest are from the natural soil cover, flooded during the construction of the Reservoir (1975). The remaining samples should contain levels of the elements that correspond to the background concentrations of these elements before they are anthropogenically affected. This was the basis for the analysis of the elements’ core depth profiles in the present study. It is noteworthy that the concentrations of most of the studied elements increase, except for Bi, La, Ti, Tl, where the concentrations decrease over time, while Ce, Gd, Th practically do not change.

The PCA results divided the analyzed chemical elements into three groups. The elements with significant impact in the formation of the first principal component (Mn, Fe, Cr, Ni, Cu, Mo, Sn, Sb and Co) have an increasing time trend in Struma sediment core (sampling site 1) and no trend (Co, Cr, Mn, Mo) or a decreasing one (Cu, Fe, Ni, Sb, Sn) at the Pchelina sampling site. The factor scores of the layers in the Pchelina sediment core (sampling site 3) increase until the late 1990s and at the end of the investigated period decreased to the levels between 1988 and 1994. The absence of an increasing trend in concentrations of the abovementioned elements at the reference point for Pchelina Reservoir (between the inflows of the two rivers—Struma River—anthropogenically affected and Svetlia River—anthropogenically unaffected) could be explained by the changed profile of the industry in the town of Pernik during the 1990s, since when mining and metallurgy have not been so dominant. These observations are supported by the calculated EFs and geoaccumulation indexes for the top layer at the Pchelina sampling site where only for Cu moderate enrichment is observed. 

The chemical elements associated with the second principal component (Ce, Gd, La, Th, and Ti) have decreasing time trends in both sediment cores at the two river inflows and excluding Th in the Pchelina sediment core too. These elements have no anthropogenic origin and the moderate enrichment of Ce and Th at the Pchelina sampling site could be attributed to their geochemical immobility [39]. The third group of chemical elements (Zn, Pb, Cd, Bi and U_nat_) forming the third principal component have significant increasing trends in Struma and Pchelina sediment cores. This leads to the formation of two distinct groups of layers in the Pchelina sediment core before and after the 1990s (layers 11 and 12). The increased concentrations of these elements at the end of the investigated period have led to moderate enrichments for Pb and U_nat_, and significant enrichments for Zn and Cd at the Pchelina sampling site. The respected geoaccumulation indexes confirm these observations with uncontaminated to moderately contaminated values for Pb and U_nat_, and moderately contaminated ones for Zn and Cd. The results from the present study largely confirm the conclusions made by Meuser and co-authors in 2006 [40] where the impact of industry located in the region of Pernik town results in increased concentrations of Pb, Cd, Cr, Cu and Zn.

The results of the conducted biotest Phytotoxkit F ™ reveal the low ecotoxicity of surface sediments, which is an indication that the concentrations of the elements classifying the Pchelina Reservoir samples as moderately contaminated has no significant ecotoxicological effect.

## 4. Materials and Methods

### 4.1. Sampling

The sampling of the bottom sediment materials was carried out in the period July–September 2020. Three locations in the Pchelina Reservoir were selected (Figure 5). Sampling point 1 is located at the inflow of the Struma River into the Pchelina Reservoir. Sampling point 2 is located at the inflow of the Svetlia River into the Pchelina Reservoir. The sampling point in Pchelina Reservoir—at the village of Radibosh (sampling point 3)—was chosen between the inflows of the two rivers—Struma River (anthropogenically affected by the industry located in the region of Pernik town) and Svetlia River (anthropogenically unaffected)—as a reference point.

Thin-walled tubes with small diameters, which ensures mechanical immersion in the sediment’s mass, were used for obtaining semi-intact specimens of fine-grained sediment samples. The basic components of tube-type samplers include steel grade S355 hardened cutting tip, body tube or barrel, and a threaded end. Pipes with a diameter of 48 × 1 mm were used, coupled to a nozzle to reach a depth of approximately 1.20 m (4 ft) below the water surface. The entire Shelby tube system follows the design requirements of ASTM D1587/D1587M–15 [41].

Sampling was carried out with a percussion-swirling motion by hand. Separate sampling tubes were packed and marked on-site and transported to the laboratory at 4 °C.

### 4.2. Sediment Digestion

Sediment samples were first pre-grinded, sieved through 2-mm sieves and homogenized. Sub-samples of 0.25 g were accurately weighed using an analytical balance, 10 mL of conc. Hydrofluoric acid (HF, 47–51%, Fisher Chemicals, Waltham, MA, USA, Trace Metal Grade) was added and the mixtures were left for 24 h. Subsequent dissolution was performed using a sand bath after adding an additional amount of 10 mL conc. HF (47–51%, Fisher Chemicals, Waltham, MA, USA, Trace Metal Grade) and 5 mL conc. Perchloric acid (HClO_4_, 70% Fisher Chemicals, Waltham, MA, USA, Trace Metal Grade). The samples were heated until the acid mixture was reduced to 1/3 of the initial volume. Then portions of 10 mL conc. HF were added and heating in a sand bath was performed until complete dissolution of the sediments. Then two portions of 10 mL conc. Nitric acid (HNO_3_, 67–69%, Fisher Chemicals, Waltham, MA, USA, Trace Metal Grade) were added and the samples were heated in a sand bath until the volume was reduced to 0.5–1 mL. After cooling, the samples were quantitatively transferred to 50 mL polyethylene tubes by repeated washing with double deionized water. All samples were initially diluted to 50 mL, and immediately before instrumental measurement, an additional dilution of 1 mL to 14 mL was performed.

### 4.3. Sediment Analysis

#### 4.3.1. Inductively Coupled Plasma Mass Spectrometry (ICP-MS)

The sediment samples were analyzed using Perkin-Elmer SCIEX Elan DRC-e ICP-MS (MDS Inc., Concord, ON, Canada) with cross-flow nebulizer. The spectrometer was optimized (RF power, gas flow, lens voltage) to provide minimal values of the ratios CeO^+^/Ce^+^ and Ba^2+^/Ba^+^ as well as maximum intensity of the analytes. The concentrations of 20 elements (Bi, Cd, Ce, Co, Cr, Cu, Fe, Gd, La, Mn, Mo, Ni, Pb, Sb, Sn, Th, Ti, Tl, U_nat_ and Zn) were determined. Coefficients for the reduction of the analytical signal (RPa, Dynamic Bandpass Tuning parameter), pre-optimized for sediment matrix, were used for the determination of Fe, Mn, and Ti using ICP-MS. They are presented in Table 5. 

Chemical elements were determined under standard conditions. In the course of the analysis, the appropriate isotopes of the elements in terms of natural distribution and low spectral interference were selected. External calibration by matrix-matched standard solutions was performed. Single element standard solutions of Bi, Cd, Ce, Co, Cr, Cu, Fe, Gd, La, Mn, Mo, Ni, Pb, Sb, Sn, Th, Ti, Tl, U_nat_ and Zn (Fluka, Steinheim, Switzerland) with initial concentration of 1000 mg/L were used to construct the calibration curve after appropriate dilution. The multielement calibration standard solutions were prepared in the concentration range from 0.2 to 20 mg/L for Fe, Mn and Ti, and in the range 0.01 to 100 µg/L for the other elements. All standard solutions were prepared with double deionized water (Millipore purification system Synergy, Molstheim, France). The calibration coefficients for all calibration curves were at least 0.99. The linearity was proven to be four orders of magnitude for Fe, Mn, and Ti, and five orders of magnitude for the other elements. The operating conditions of the ICP-MS are listed in Table 6.

To verify the accuracy of the analysis, stream sediment certified reference materials STSD-1 and STSD-3 of (Canada Center for Mineral and Energy Technology, Geological Survey of Canada) were subjected to analysis using the same sample preparation. Table 7 and Table 8 present the experimentally determined and the certified values of the analyzed elements in the certified reference materials STSD-1 and STSD-3, respectively. If the values of the analytical yields are in the range of 97–108%, the method is considered fit-for-purpose. All measurements were performed in triplicate and the mean value was reported.

#### 4.3.2. Gamma-Spectrometry

The individual sediment samples were air-dried and cleaned from plant impurities (roots, leaves, shells, etc.), followed by sieving through 2-mm sieves and homogenization. Samples of about 10 g were packed in standard geometry vessels and measured by gamma-spectrometry for the determination of the total activity of ^137^Cs. The activity of radionuclides was measured using HPGe detector Canberra 7229 (energy resolution 1.8 and efficiency 16% at 1332.5 keV) coupled to a 4196-channel analyzer Canberra 35Plus. The calibration was achieved using national standard radioactive solutions and standard samples, produced, and standardized at Czech Metrology Institute (Serial No.: 130520-1785043). The accuracy and precision of the analysis have been verified by participation in International Atomic Energy Agency (IAEA) round robin tests. All measurements were performed in triplicate and the mean value was reported.

#### 4.3.3. Ecotoxicological Studies

The ecotoxicological test Phytotoxkit F ™ [42] measures the reduction of seed germination (SG) and the growth of young roots (RG) of selected higher plants (Sorghum saccharatum, Lepidium sativum and Sinapis alba) when seeded in contaminated samples for several days compared to a control sample.

Sample preparation included drying, grinding, and sieving through 2-mm sieves as a preliminary step. Then the Phytotoxkit F ™ tests were performed on water-saturated samples. It was experimentally found that 35 mL of distilled water was required to achieve 100% saturation of control soil with a volume of 90 cm^3^. The determination of the water retention capacity of the analyzed sediments was performed on a representative sample (mean of the three analyzed sediments). To 60 mL of distilled water, 90 cm^3^ of the sample was added and after equilibrium was reached, the volume of excess water (15 mL) was measured, and the volume of water required to achieve saturation of the sample was calculated (45 mL).

The first step of conducting the Phytotoxkit F ™ test was to place 90 cm^3^ of each of the studied sediments, as well as of the control sample, in special plastic test plates. This was followed by hydration with the necessary volume of distilled water to achieve saturation, placement of black filter paper, seeding of 10 seeds of the test plant (Sinapis alba—white mustard) and closing the test plate. Two repetitions for each sediment sample were made, and 3 for the control sample—Reference OECD soil for Phytotoxkit test (Microbiotests, Gent, Belgium). The test plates were placed vertically in an incubator for 72 h at a temperature of 25 ± 1 °C in the dark. As the last step, the number of germinated seeds was counted and the length of the roots was measured, using the free software ImageJ [43].

The ecotoxicological effect (%), reflecting the germination of the seeds is calculated by 100 × (A − B)/A, where A is the average number of germinated seeds for the control sample and B is the average number of germinated seeds for the analyzed sample.

The ecotoxicological effect (%), reflecting the growth of the roots, is calculated by 100 × (A − B)/A, where A is the mean length (mm) of the plant roots from the control sample and B is the mean length (mm) of the plant roots from the sample.

### 4.4. Statistical Analysis

#### 4.4.1. Principal Component Analysis (PCA)

The Principal Component Analysis (PCA) is a multivariate approach to data reduction. The aim is to find and interpret the latent interdependencies between the variables (chemical elements) in the data set. Such variables form new ones, called latent factors or principal components. In addition to discovering the data structure, the PCA data set can be modelled, compressed, classified and visualized on a plane. The main task in PCA is the decomposition of the data matrix into two parts—a matrix of factor results and a matrix of factor weights. The factor weights show the participation of each of the old variables in the formation of the main components while the factor results are the coordinates of the objects (layers in the sedimentary cores) in the newly formed variables. The determination of the number of significant principal components is based on their eigenvalues and the percentage of explained variation in the data.

#### 4.4.2. Mann–Kendall Test

The Mann–Kendall test is a non-parametric approach for estimating time trends. The assessment uses all possible discrepancies between the values for a given layer of sediment with those of previous years. Positive differences (increase in the concentration of the element) are marked with +1, negative (decrease in the concentration of the element) with −1, and the lack of difference with 0. The test takes into account only the sign of the differences. The null hypothesis is that there is no time trend, the alternative hypothesis is that there is a positive or negative time trend. The direction of the trend is determined by the sign of the S statistics, which is the difference between the number of positive and negative differences. At the assumed significance level (α = 0.05), the null hypothesis is rejected at values of *p* < 0.05.

#### 4.4.3. Enrichment Factor (EF)

To distinguish anthropogenic pollution from the natural content of elements in the sediment, enrichment factors (EF) were calculated by comparing the measured concentrations of chemical elements with the geochemical background values of the study area. To avoid erroneous enrichment results, geochemical normalization based on the concentration of a conservative element is usually used. The purpose of normalization is to correct changes in the nature of the sediment that may affect the distribution of contaminants. Al, Fe, Th, Ti and Zr are usually used as conservative elements [15,44]. The normalized enrichment factor (EF) is determined by the metal/X concentration ratio (X = Fe, Al, Th, Ti, Zr) divided by the background metal concentration/background concentration ratio X:(1)$$EF=\frac{{\left(\frac{Me}{X}\right)}_{sample}}{{\left(\frac{Me}{X}\right)}_{background}},$$

Five levels of pollution are often identified—EF < 2: low enrichment; EF 2–5: moderate enrichment; EF 5–20: significant enrichment; EF 20–40: strong enrichment; EF > 40: extremely strong enrichment. In addition, values of 0.5 ≤ EF ≤ 1.5 suggest that the concentration of the elements may come entirely from natural weathering processes. However, EF > 1.5 shows that a significant part of the microelements did not come from the earth’s crust, i.e., their origin is from other sources, such as point and non-point pollution and biota [17].

#### 4.4.4. Geoaccumulation Index (*I_geo_*)

A similar approach was used for the determination of the Geoaccumulation Index (*I_geo_*):(2)$$I_{geo}=log_{2}\left(\frac{C_{n}}{1.5C_{ref}}\right),$$
where *C_n_* is the concentration of the element in the sample and *C_ref_* is the background concentration [45].

The coverage factor of 1.5 used allows the normalization of possible variations in the data for background concentrations, which may also be due to anthropogenic pollution. 7 classes of contamination are known depending on the Geoaccumulation Index (*I_geo_* ≤ 0—uncontaminated sample; 0 < *I_geo_* < 1 uncontaminated to moderately contaminated sample; 1 < *I_geo_* < 2 moderately contaminated sample; 2 < *I_geo_* < 3 moderately to highly contaminated sample; 3 < *I_geo_* < 4 heavily contaminated sample; 4 < *I_geo_* < 5 heavily to extremely contaminated sample; *I_geo_* > 5 extremely contaminated sample) [14,46,47,48]. The highest class corresponds to at least a 100-fold difference with the background concentration.

## 5. Conclusions

Based on the measurement of γ-activity of the technogenic ^137^Cs, an accumulation of an average of 15 cm of sediment was established for 34 years, at a sedimentation rate of an average of 0.44 cm/year.

The distribution of the concentrations of chemical elements in the sediment from the three sampling points (1—Pchelina Reservoir at the flow of Struma River, 2—Pchelina Reservoir at the flow of Svetlia River and 3—Pchelina Reservoir near the village of Radibosh) is presented. PCA of the data shows three main components, which describe nearly 80% of the variation of the data. The first main component (41.53% of the data variation) contains Mn, Fe, Cr, Ni, Cu, Mo, Sn, Sb and Co. The factor scores show that the concentrations of these elements decrease in the order Pchelina > Struma > Svetlia. All these elements have a positive time trend in the Struma sediment core, which is an indication that most of the elements in the Reservoir come through the anthropogenically affected river Struma. The second main component (22.16%) is formed by Ce, Gd, La, Th, and Ti, which decrease significantly with time in Svetlia and Struma. The third main component (15.44%) is formed by the elements Zn, Pb, Cd, Bi and U_nat_. The factor scores of the layers in the sedimentary cores show the anthropogenic origin of most of these elements. There is an increase over time in the sedimentary cores in Struma and Pchelina. The increase in the sedimentary layers of Pchelina is especially pronounced. In the first group (beginning of the studied period), there are contents comparable to the sediment core of Svetlia, while the contents of the elements in the second group are the highest for the three studied sediment cores.

To distinguish anthropogenic pollution from the natural content of elements in the sediment, enrichment factors (EF) have been calculated for which Fe concentrations have been used as a conservative element. Significant enrichment was observed only at the reference point (Pchelina), for Zn and Cd. In terms of Cd, the enrichment in Svetlia is moderate. The enrichment factor is 1.8 in the Struma, which shows significant contamination with this metal in the entire Reservoir. Except for these elements, moderate enrichment is observed for Cu, Pb, Ce and Th only in Pchelina, as well as for T1 and U_na_t in Svetlia and Pchelina. For most elements, the geoaccumulation index corresponds to an uncontaminated sample. In Svetlia, only for Cd, T1 and U_nat_ the index corresponds to an uncontaminated to moderately contaminated sample. The index of geoaccumulation in the sediments of Struma corresponds to an uncontaminated to moderately contaminated sample for Fe, Cr, Mo and Sn and to a moderately contaminated sample for Sb and Cd. The geoaccumulation index for the sediments of Pchelina corresponds to an unpolluted to moderately contaminated sample for Pb, Tl and U_nat_ (similar to the Svetlia sediments) and a moderately polluted sample for Zn and Cd (similar to the Svetlia sediments). Moderate contamination of the sediments of Pchelina Reservoir in all sampling points is from Cd.

The results of the Phytotoxkit F bioassay revealed the low ecotoxicity of Pchelina Reservoir surface sediments in terms of both seed germination (SG) and root growth (RG) of the plant species Sinapis alba.

## Figures and Tables

**Figure 1 molecules-26-07517-f001:**
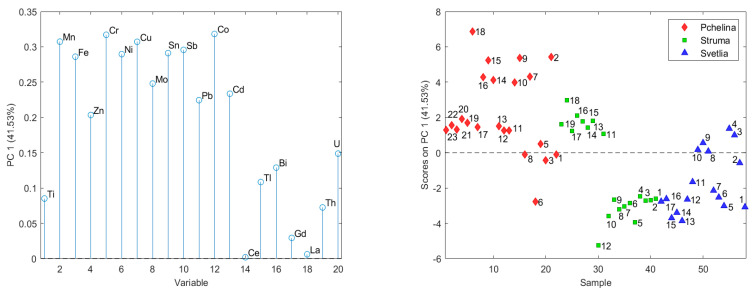
Factor weights and factor scores of the first principal components (the numbers on the graph indicate the historical sequence of layers in sediment core with the first (1) being the oldest).

**Figure 2 molecules-26-07517-f002:**
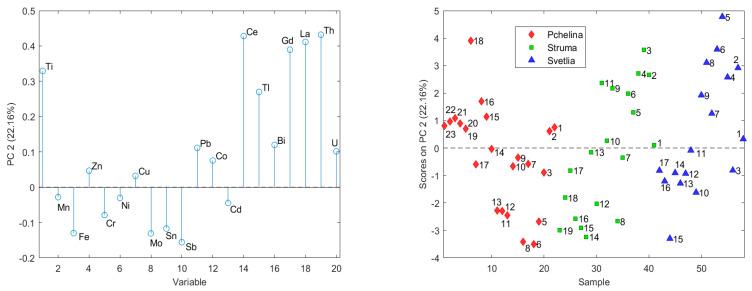
Factor weights and factor scores of the second principal components (the numbers on the graph indicate the historical sequence of layers in sediment core with the first (1) being the oldest).

**Figure 3 molecules-26-07517-f003:**
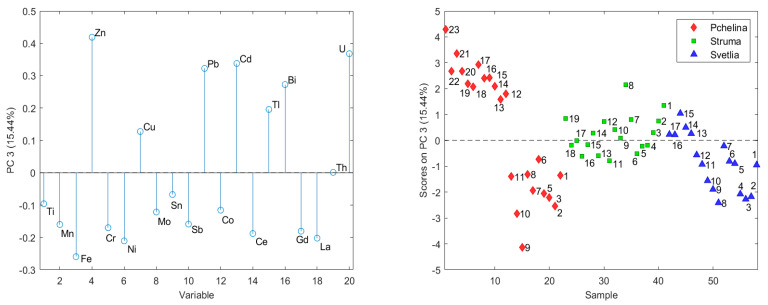
Factor weights and factor scores of the third principal components (the numbers on the graph indicate the historical sequence of layers in sediment core with the first (1) being the oldest).

**Figure 4 molecules-26-07517-f004:**
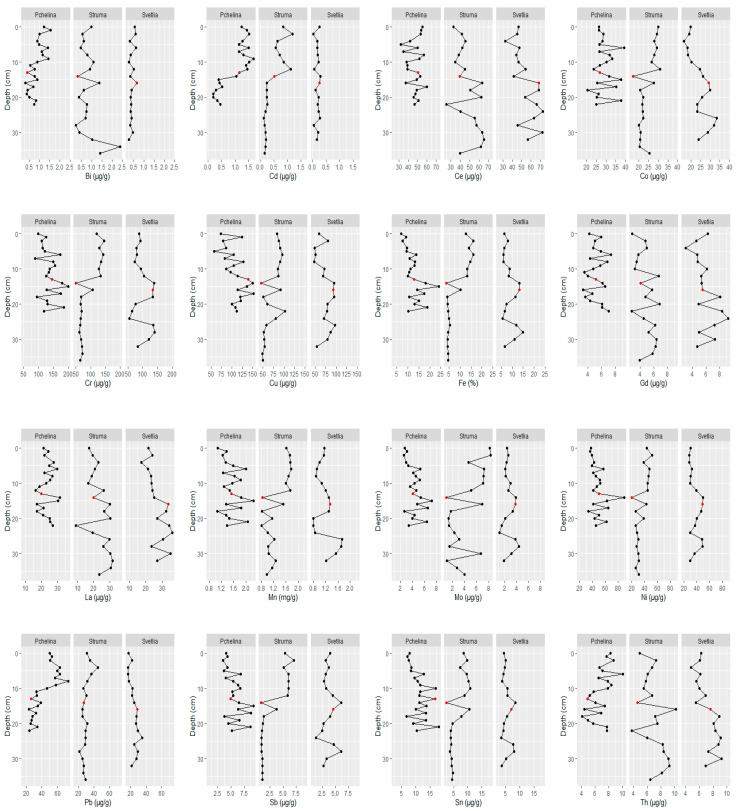
Elements’ depth profiles of sediment cores taken at Pchelina, Struma and Svetlia.

**Figure 5 molecules-26-07517-f005:**
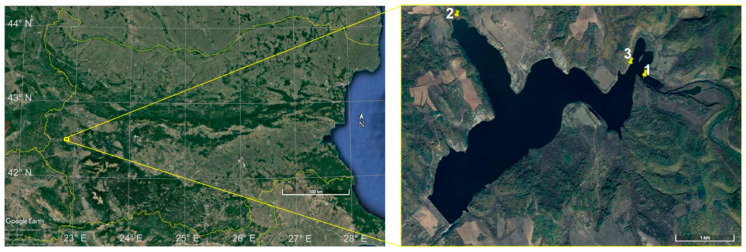
Sampling locations (1-Struma, 2-Svetlia and 3-Pchelina).

**Table 1 molecules-26-07517-t001:** Trend analysis of sediment cores at three sampling points.

Element	Pchelina	Struma	Svetlia
Bi	+		
Cd	+	+	
Ce		–	–
Co		+	–
Cr		+	
Cu	–	+	
Fe	–	+	
Gd			–
La		–	–
Mn		+	
Mo		+	
Ni	–	+	
Pb	+	+	–
Sb	–	+	
Sn	–	+	
Th	+		–
Ti		–	–
Tl	+		–
U_nat_	+	+	+
Zn	+	+	

**Table 2 molecules-26-07517-t002:** Enrichment factor (EF) in the three sampling points (yellow—moderate enrichment, red—significant enrichment).

Element	EF Struma River	EF Svetlia River	EF Pchelina
Ti	0.3	1.2	1.8
Mn	0.5	1.3	1.3
Fe	–	–	–
Zn	0.7	1.8	** 7.6 **
Cr	0.7	1.3	1.4
Ni	0.5	1.4	1.5
Cu	0.5	1.4	** 2.4 **
Mo	1.0	1.3	1.2
Sn	0.7	1.4	1.3
Sb	** 2.0 **	1.7	1.3
Pb	0.4	1.2	** 3.6 **
Co	0.5	1.2	1.8
Cd	1.8	** 3.2 **	** 7.5 **
La	0.2	1.4	1.7
Ce	0.2	1.4	** 2.4 **
Tl	0.4	** 2.7 **	** 3.5 **
Bi	0.4	1.9	1.3
Gd	0.2	1.3	1.7
Th	0.2	1.4	** 2.3 **
U_nat_	0.4	** 2.8 **	** 3.5 **

**Table 3 molecules-26-07517-t003:** Geoaccumulation index (*I_geo_*) in the three sampling points (yellow—uncontaminated to moderately contaminated sample, red—moderately contaminated sample).

Element	*I_geo_* Struma	*I_geo_* Svetlia	*I_geo_* Pchelina
Ti	−1.01	−1.06	−0.70
Mn	−0.03	−0.95	−1.19
Fe	** 0.85 **	−1.37	−1.58
Zn	** 0.32 **	−0.55	** 1.35 **
Cr	** 0.26 **	−0.97	−1.13
Ni	−0.01	−0.91	−1.00
Cu	−0.02	−0.90	−0.31
Mo	** 0.84 **	−0.98	−1.35
Sn	** 0.44 **	−0.93	−1.26
Sb	** 1.87 **	−0.61	−1.18
Pb	−0.36	−1.15	** 0.27 **
Co	−0.11	−1.10	−0.74
Cd	** 1.69 **	** 0.30 **	** 1.33 **
La	−1.24	−0.90	−0.79
Ce	−1.34	−0.88	−0.34
Tl	−0.58	** 0.06 **	** 0.25 **
Bi	−0.61	−0.44	−1.19
Gd	−1.51	−0.99	−0.85
Th	−1.28	−0.91	−0.37
U_nat_	−0.33	** 0.14 **	** 0.21 **

**Table 4 molecules-26-07517-t004:** Ecotoxicological studies using Phytotoxkit F ™ applied to the sediments from the three sampling points.

Sampling Point	Ecotoxicological Effect SG (%)	Ecotoxicological Effect RG (%)
Struma	0	23.26
Svetlia	0	17.56
Pchelina	0	−0.26

**Table 5 molecules-26-07517-t005:** RPа coefficients used for the determination of Fe, Mn and Ti, in sediment samples.

Isotope	RPa Coefficient
^47,49,50^Ti	0.012
^54^Fe	0.014
^56^Fe	0.016
^57^Fe	0.013
^55^Mn	0.014

**Table 6 molecules-26-07517-t006:** ICP-MS instrumental conditions for the determination of chemical elements in sediment samples.

Instrument	Operating Conditions
Cooling Ar gas flow	15 L/min
Auxiliary Ar gas flow	1.20 L/min
Nebulizer gas flow	0.85 L/min
Lens voltage	6.00 V
ICP RF power	1100 W
Integration time	2000 ms
Dwell time	50 ms
Acquisition mode	Peak hop
Sample uptake rate	2 mL/min
Rinse time	180s
Rinsing solution	3% HNO_3_

**Table 7 molecules-26-07517-t007:** Measured and certified values of the sediment certified reference material STSD-1.

Element	Isotope	LOQ (μg/g)	Measured Value in STSD-1 (μg/g)	Certified Value (μg/g)	Recovery %
Ti	46	1.28	4632	4600	100.7
Mn	55	0.98	3918	3950	99.2
Fe %	57	1.04	4.73	4.7	100.6
Zn	64	0.17	174	178	97.8
Cr	52	0.17	65.9	67	98.4
Ni	62	0.06	25.3	24	105.4
Cu	65	0.12	37.4	36	103.8
Mo	95	0.07	1.37	<5	–
Sn	116	0.22	4.13	4	103.3
Sb	123	0.07	3.42	3.3	103.6
Pb	208	0.08	35.6	35	101.9
Co	59	0.03	17.4	17	102.6
Cd	112	0.02	1.39	–	–
Ce	140	0.03	50.1	51	98.2
Tl	205	0.02	0.45	–	–
Bi	209	0.02	0.69	–	–
Gd	158	0.03	5.6	–	–
La	139	0.03	29.2	30	97.3
Th	232	0.2	3.83	3.7	103.5
U_nat_	238	0.2	8.0	8	100.4

**Table 8 molecules-26-07517-t008:** Measured and certified values of the sediment certified reference material STSD-3.

Element	Isotope	LOQ (μg/g)	Measured Value in STSD-3 (μg/g)	Certified Value (μg/g)	Recovery %
Ti	46	1.28	4370	4400	99.3
Mn	55	0.98	2763	2730	101.2
Fe %	57	1.04	4.49	4.4	102.0
Zn	64	0.17	214	204	105.1
Cr	52	0.17	82.5	80	103.1
Ni	62	0.06	29.2	30	97.3
Cu	65	0.12	38.8	39	99.4
Mo	95	0.07	5.92	6	98.6
Sn	116	0.22	4.31	4	107.6
Sb	123	0.07	4.16	4	104.0
Pb	208	0.08	41.8	40	104.4
Co	59	0.03	16.15	16	100.9
Cd	112	0.02	1.09	–	–
Ce	140	0.03	62	63	98.5
Tl	205	0.02	0.42	–	–
Bi	209	0.02	0.45	–	–
Gd	158	0.03	5.2	–	–
La	139	0.03	38.4	39	98.5
Th	232	0.2	8.6	8.5	101.5
U_nat_	238	0.2	10.3	10.5	97.6

## Data Availability

The data presented in this study are available on request from the corresponding author.

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
