# Peer review of "Sediment Assessment of the Pchelina Reservoir, Bulgaria"

_molecules, 2021, doi:10.3390/molecules26247517_

Round 1

Reviewer 1 Report

This manuscript addresses an important topic about temporal sediment contamination of the Pchelina Reservoir, Bulgaria; and has done a very thorough analytical approach to the issue. As there is apparently little published about this area, the data have some value. However, the manuscript could be improved.

My comments and suggestions are the following:

  1. The authors used different terms to refer to a group of elements: Metal(loid)s, metals, elements, toxic metals, heavy metals. Please see (Pourret and Hursthouse 2019; Pourret et al. 2021). I do not recommend using a specific term, please unify the used terms.
  2. Page 1 Lines 38 and 44; Please paraphrase these sentences as they are a repetition of the same facts.
  3. Page 2 Line 64; ‘‘thus very slowly turning into a rock’’??? Please remove this sentence. If the sediment is buried deeply (thousands of feet), it becomes compacted and cemented, forming sedimentary rock.
  4. Page 2 Lines 65:72; these sentences need some appropriate references.
  5. Page 2 Line 88; Pchelina Dam or Pchelina reservoir.
  6. Page 3 Line 103; ‘‘This component gives a general idea of the chemical composition of the sediments’’ Explain a bit more.
  7. Page 6 Line 149; ‘‘below the sample’’ change to below the surface of the sample.
  8. Page 7 Lines 158:159; ‘‘The enrichment factor is 8 in the Struma, which shows significant pollution with this metal’’ Is this correct??
  9. Considering the calculated EF, Struma River (sampling point 1) shows the lowest values except for Sb; the authors stated in many places that the Struma River is the main pollutant for the Pchelina reservoir. In addition, when comparing the calculated EF values with those of Igeo values Struma River (sampling point 1) show the highest values for most of the investigated elements. The authors used the same reference concentration for EF and Igeo It should give similar results. Please revise the calculated values or explain these differences.
  10. Page 9 Lines 232:233; ‘‘moderate enrichment of Ce and Th at the Pchelina sampling site could be attributed to their geochemical behaviour’’ Explain a bit more.
  11. Page 9 Figure 5; This figure needs a coordinate and a clearer scale.
  12. Page 10 Lines 271:284; these sentences need some appropriate references.
  13. Page 13 Lines 386:388; these sentences need some appropriate references.
  14. Page 13 Line 390; Please refer to Müller (1969), it’s well known that it was adopted by Müller (1969).
  • Pourret O, Hursthouse A (2019) It’s Time to Replace the Term ’’Heavy Metals‘‘ with ’’Potentially Toxic Elements‘‘ When Reporting Environmental Research. Int J Environ Res Public Health 16 (22). https://doi.org/10.3390/ijerph16224446
  • Pourret O, Bollinger J, Hursthouse A (2019) Heavy metal: a misused term?. Acta Geochim (2021) 40(3):466–471. https://doi.org/10.1007/s11631-021-00468-0
  • Müller, G. 1969. Index of geoaccumulation in sediments of the Rhine River. Geo. J. 2 (3), 108–118.

Author Response

Dear Reviewer 1,

Thank you for your time and efforts to review our paper. Please find below the responses to your questions and remarks.

Kind regards,

The authors

Reviewer 2 Report

The present manuscript entitled Sediment Assessment of the Pchelina Reservoir, Bulgaria by Venelinov et al. (molecules-1487286) describes studies on temporal dynamics of anthropogenic impacts on the Pchelina Reservoir. The chemical element analysis of three sediment cores was performed using ICP-MS and gamma spectrometry (analysis of 137Cs for calculation of sedimentation rate). The obtained results were first analyzed using PCA and then by the Mann-Kendall test for estimating time trends. Enrichment factors and geoaccumulation indexes were calculated and used to interpret the obtained data. Based on the Phytotoxkit F bioassay, the low ecotoxicity of Pchelina reservoir sediments was proved.

The present article is well written, contains properly prepared figures, and has good structure. The conducted studies have been described in detail. The paper meets Molecules' requirements, and I recommend the article for publication in Molecules following the common editing stage. My current decision is a minor revision. More specific comments and observations are presented below.

  1. Information about methods used for chemical element analysis can be added to the abstract.
  2. The article lacks a well-conducted validation of the analytical methods. If possible, I would ask you to calculate basing and missing validation parameters. The method validation parameters can be determined based on the constructed calibration curves. Information about LOD, LOQ, linear range can be added to the abstract.
  3. I would like to ask how the authors predict further contamination of the Pchelina reservoir?
  4. Numeric values presented in tables and some in the text contain a "comma", but there should be a "dot" i.e., in Table 2, Table 3, Table 4, and others.
  5. Section 4.1. I would like a clear description of the place corresponding to the reference point. Could the authors mark this point in Figure 5? How many samples were taken from each location?
  6. Section 4.3. What was the size of the sieves? It would be appropriate to add an explanation to the abbreviation "HF". The authors write that they used polyethylene tubes to prepare the solutions. Was it not better to use volumetric flasks?
  7. Generally Materials and Methods section. Please remember to add not only the company name but also the country of origin for reagents and instrumentation. A good description of the reagents used would be valuable for properly understanding the tests performed.
  8. Section 4.3.1. The authors wrote about low spectral interferences. What can be done in the event of strong interference effects? How would you deal with them? What types of interference effects could occur? Could you add more information about multi-element standard (composition, concentration, in which solution, company)? What was the range of concentrations used for the calibration curve? In what solvent were the solutions prepared? Can calibration curves be added to the article? On their basis, the validation of the analytical procedure can be performed.
  9. Description of Table 6. Elements should be instead of an element.
  10. Please improve the index in HNO3 at the end of Table 6.
  11. The authors present the recovery values in Tables 7 and 8. I would like to ask what the limitations in calculating the recovery based on CRM are? In the description of Table 8 should be STSD-3.
  12. Section 4.3.2. Could you add more information about standard radioactive solutions and standard samples? Would you please add an explanation of IAEA?
  13. Section 4.3.3. What were the parameters of water used?
  14. What software was used to analyze the data?
  15. Author Contributions. Information about software appears two times.
  16. There are also typos in the prepared paper, and they should be corrected, i.e., Page 9, line 239: Index in Unat. Page 13, line 392, concentrations should not be in capital letters.
  17. I also observed a few typos in references. In [10], the DOI number is most likely wrong.

I hope that the comments presented will help improve the article

Author Response

Dear Reviewer 2,

Thank you for your time and efforts to review our paper. Please find below the responses to your questions and remarks.

Kind regards,

The authors
